# Atmospheric Mercury Deposition in Macedonia from 2002 to 2015 Determined Using the Moss Biomonitoring Technique

**Trajče Stafilov [1,\*], Lambe Barandovski [2,\*], Robert Šajn [3] and Katerina Bačeva Andonovska [4]**

[1] Institute of Chemistry, Faculty of Natural Sciences and Mathematics, Ss. Cyril and Methodius University, POB 162, 1000 Skopje, Macedonia

[2] Institute of Physics, Faculty of Natural Sciences and Mathematics, Ss. Cyril and Methodius University, POB 162, 1000 Skopje, Macedonia

[3] Geological Survey of Slovenia, Dimičeva 14, 1000 Ljubljana, Slovenia; Robert.Sajn@GEO-ZS.SI

[4] Research Center for Environment and Materials, Academy of Sciences and Arts of the Republic of North Macedonia—MANU, Krste Misirkov 2, 1000 Skopje, Macedonia; kbaceva@manu.edu.mk

\* Correspondence: trajcest@pmf.ukim.mk (T.S.); lambe@pmf.ukim.mk (L.B.);
Tel.: +38970350756 (T.S.); +38970607921 (L.B.)

**Abstract:** The moss biomonitoring technique was used in 2002, 2005, 2010 and 2015 in a potentially toxic elements study (PTEs) in Macedonia. For that purpose, more than 70 moss samples from two dominant species (*Hypnum cupressiforme* and *Homalothecium lutescens)* were collected during the summers of the mentioned years. Total digestion of the samples was done using a microwave digestion system, whilst mercury was analyzed by cold vapour atomic absorption spectrometry (CV–AAS). The content of mercury ranged from 0.018 mg/kg to 0.26 mg/kg in 2002, from 0.010 mg/kg to 0.42 mg/kg in 2005, from 0.010 mg/kg to 0.60 mg/kg in 2010 and from 0.020 mg/kg to 0.25 mg/kg in 2015. Analysis of the median values shows the increase of the content in the period 2002–2010 and a slight reduction of the air pollution with Hg in the period 2010–2015. Mercury distribution maps show that sites with increased concentrations of mercury in moss are likely impacted by anthropogenic pollution. The results were compared to similar studies done during the same years in neighboring countries and in Norway—which is a pristine area and serves as a reference, and it was concluded that mercury air pollution in Macedonia is significant primarily in industrialized regions.

**Keywords:** air pollution; moss biomonitoring; potentially toxic elements; mercury; CV–AAS; Macedonia

## 1. Introduction

Air pollutants are considered all chemical compounds or elements released into the atmosphere that pose health hazards to ecosystems and humans [1]. The majority of potentially toxic elements (PTEs) originate mainly from anthropogenic sources [2–4]. Natural sources of these elements include volcanoes, forest fires, biological decomposition processes and oceans [4]. The largest anthropogenic sources of PTEs in the atmosphere are the combustion of fossil and biofuels, traffic and emissions from industrial processes [5,6].

Due to the specific features and the effects on human health, the discharge of mercury in the environment has been identified as a global problem [7]. Even the mercury is naturally occurring in the Earth's crust, the atmospheric emission is mostly in an elemental mercury vapour form. Mercury enters the environment through volcanic eruptions and erosion of natural mercury–containing deposits, but also from forest fires and uncontrolled coal bed fires [7]. Anthropogenic sources of Hg are

mostly connected with extraction refining and use of fossil fuels, metal production (Cu, Zn, Pb, etc.), production of inorganic materials (cement, paper production), recycling processes, chlorine–alkaline processes, etc. [7] High concentrations of mercury in human organism impact the central nervous system, especially the sensory, auditory and visual, parts of the brain that can affect coordination, lower immunity, heart attack risk, nervous system damage, and impair reproduction [8,9]. Therefore, timely and reliable identification of mercury discharge and presence in the environment is crucial.

Estimation of atmospheric heavy metal deposition using carpet–forming moss was done for the first time in the 1960s by Rühling and Tyler [10]. The moss analysis technique provides an alternative, time–integrated measure of the spatial patterns of PTEs deposition from the atmosphere to terrestrial ecosystems. The technique avoids the need for deploying large numbers of deposition collectors with an associated long–term programme of routine sample collection. Since 1990, the European moss survey has been repeated at five–yearly intervals [5,6,11,12] and the latest survey was conducted in 2015 with more than 35 participating countries [13]. The European moss survey gives data on the contents of ten potentially toxic metals (As, Cd, Cr, Cu, Fe, Hg, Ni, Pb, V, Zn) as well as the content of nitrogen [13–16].

The first study of mercury air pollution on the whole territory of Macedonia, using moss species as biomonitors, was undertaken in 2002, and the study was repeated in 2005, 2010, and 2015 within the European moss survey [17–23]. Pollution of soil samples with Hg, obtained in the vicinity of town Veles, due to work of the lead and zinc metallurgical plant situated near the town, was studied by Stafilov et al. [24,25] showing the increase of the content of Hg in topsoil over the European Hg average by a factor of 3.2. Analysis of the 174 soil samples collected in Kavadarci, an area in the south of the country [26,27] showed both lithogenic and anthropogenic influence of mercury in the collected samples. In some area along the river Vardar and the city of Kavadarci, the content of Hg was up to 3.8 mg/kg. Increased content of Hg was also found in soil samples in the vicinity of the "Allchar" As–Sb–Tl mine, located to the south of the city of Kavadarci [28]. This mine is lithogenic in origin but contributes to localized air pollution and it has been shown to influence the communities in this geographic area presenting natural phenomena.

The aims of this work are to investigate and present the temporal trends of the content of Hg in moss samples from the results of the surveys performed in the 2002–2015 period, to determine the places most affected by Hg pollution, and to try to connect the pollution with known anthropogenic activities in the regions, to distinguish natural from anthropogenic sources, to identify the deposition patterns and to compare results with previous studies in the neighboring countries and pristine areas.

## 2. Materials and Methods

### 2.1. Study Area

The Republic of Macedonia is a landlocked country situated in the central part of the Balkan Peninsula. Macedonia is bordering Serbia and Kosovo to the north, Bulgaria to the east, Albania to the west, and Greece to the south (Figure 1). A detailed description of the country (location, climate, and demographics) can be found elsewhere [17–23,29,30]. The location of main industrial activities and their input of different PTEs, has been also previously reported in several studies [31–46].

### 2.2. Sampling Sample Preparation and Instrumentation

In 2002, 2005, 2010 and 2015 moss samples from two moss species were collected on the entire territory of Macedonia [17–23]. In total, 72 moss samples were collected in each of the sampling campaigns (Figure 2). Collected moss samples were from the two most abundant moss species (*Hypnum cupressiforme* and *Homalothecium lutescens*). In locations where the two–moss species were collected, the interspecies comparison showed no differences within error estimates. The sampling procedure followed the principles of European moss surveys [12,47–49]. The moss samples were

previously digested by microwave digestion system at 180 °C with nitric acid and then mercury was determined by cold vapor atomic absorption spectrometry (CV–AAS) [50].

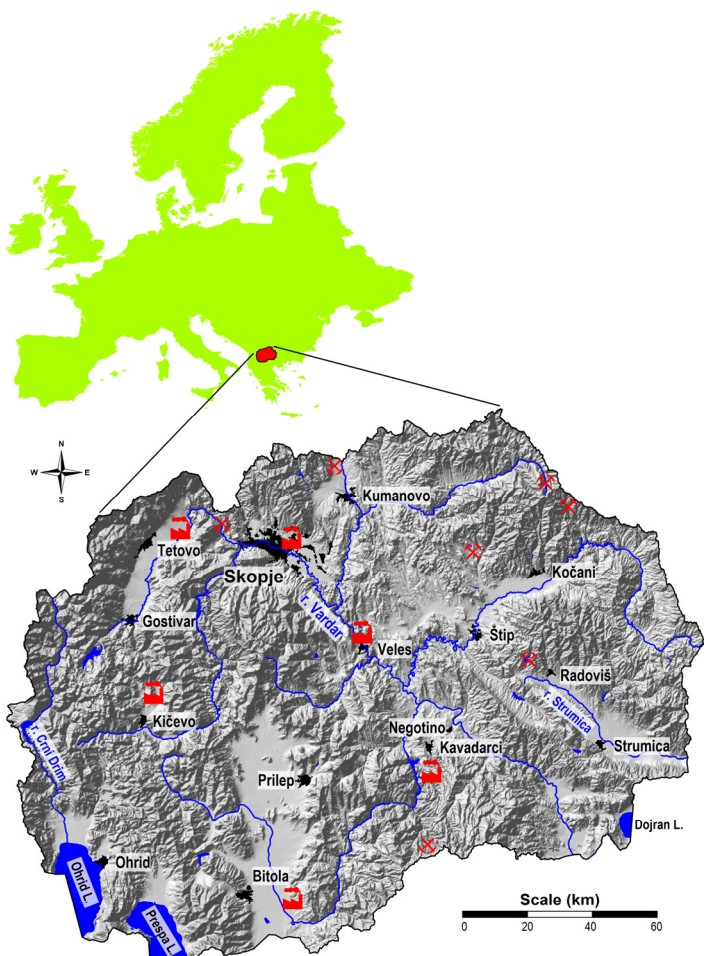

**Figure 1.** Map of the Republic of Macedonia.

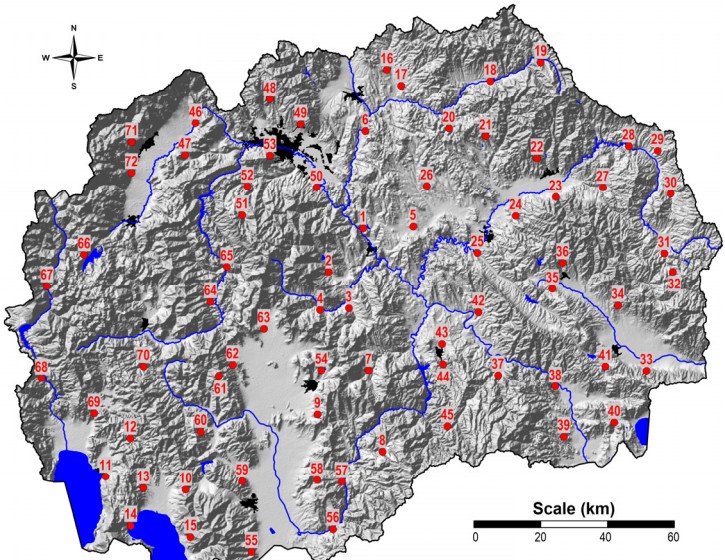

**Figure 2.** Location of sampling points.

### 2.3. Quality Control

The quality control of Hg determination was ensured by standard reference materials M2 and M3, which are prepared for the European Moss Survey [50,51] as well as the standard addition method.

### 2.4. Statistical Methods

Statistical analysis of the obtained data from the studies were made using the statistical software Statistica 13 (StatSoft, Inc., Tulsa, OK, USA) [52,53]. The common universal kriging with linear variogram interpolation method was applied to construct the maps of mercury areal distribution [54]. The basic grid cell size for interpolation was $1 \times 1$ km. For class limits the percentile values of distribution of all interpolated values (2002/2005/2010/2015) were chosen. Seven classes of the following percentile values were selected: 0–10, 10–25, 25–40, 40–60, 60–75, 75–90 and 90–100. In addition, an analysis of variance (ANOVA) was performed which showed significant differences between the four sampling seasons.

## 3. Results and Discussion

Results of the descriptive statistics of mercury content in the moss samples collected in 2002, 2005, 2010, and 2015 are given in Table 1. In Table 2, the obtained results were compared with the results of the content off Hg in moss samples obtained from similar studies in the neighbouring countries, as well as Norway which is considered a pristine area. The maps of the spatial distribution of Hg in moss samples collected in 2002, 2005, 2010, and 2015 to observe the trends of pollution in Macedonia during this period are given in Figures 3 and 4.

**Table 1.** Descriptive statistics of measurement according to sampling campaign, (in mg/kg).

| | $N$ | $X_a$ | Md | Min | Max | $P_{10}$ | $P_{90}$ | S | CV | A | E |
|---|---|---|---|---|---|---|---|---|---|---|---|
| 2002 | 72 | 0.069 | 0.056 | 0.018 | 0.26 | 0.034 | 0.114 | 0.040 | 60 | 2.03 | 7.08 |
| 2005 | 72 | 0.080 | 0.068 | 0.010 | 0.42 | 0.012 | 0.14 | 0.072 | 89 | 2.40 | 7.69 |
| 2010 | 72 | 0.110 | 0.093 | 0.010 | 0.60 | 0.05 | 0.16 | 0.094 | 84 | 3.57 | 14.2 |
| 2015 | 72 | 0.087 | 0.084 | 0.020 | 0.25 | 0.020 | 0.15 | 0.050 | 58 | 0.67 | 0.75 |

$N$—number of samples, $X_a$—arithmetic mean, Md—median, Min—minimum, Max—maximum, $P_{10}$—10 percentile, $P_{90}$—90 percentile, S—standard deviation, CV—coefficient of variation, A—asymmetry, E—distribution.

**Table 2.** The median values and ranges for the content of Hg obtained for Macedonia, neighbouring countries, and Norway.

| | No. of Samples | Median (mg/kg) | Range (mg/kg) |
|---|---|---|---|
| Macedonia, 2002 | 72 | 0.056 | 0.018–0.26 |
| Serbia, 2000 [55] | 92 | 0.386 | 0.01–2.69 |
| Norway, 2000 [11] | 464 | 0.052 | 0.022–0.208 |
| Macedonia, 2005 | 72 | 0.068 | 0.010–0.42 |
| Croatia, 2006 [50] | 94 | 0.064 | 0.007–0.301 |
| Slovenia, 2005 [48] | 57 | 0.095 | 0.050–0.175 |
| Norway, 2005 [48] | 100 | 0.046 | 0.026–0.166 |
| Macedonia, 2010 | 72 | 0.093 | 0.010–0.60 |
| Croatia, 2010 [56] | 121 | 0.043 | 0.010–0.145 |
| Slovenia, 2010 [6] | 63 | 0.056 | 0.030–0.16 |
| Albania 2010 [6] | 59 | 0.130 | 0.031–2.23 |
| Kosovo 2010 [6] | 25 | 0.033 | 0.009–0.35 |
| Norway, 2010 [6] | 463 | 0.060 | <0.024–0.34 |
| Macedonia, 2015 | 72 | 0.084 | 0.020–0.25 |
| Albania, 2015 [57] | 55 | 0.049 | 0.006–0.21 |
| Norway, 2015 [58] | 229 | 0.050 | 0.005–0.53 |

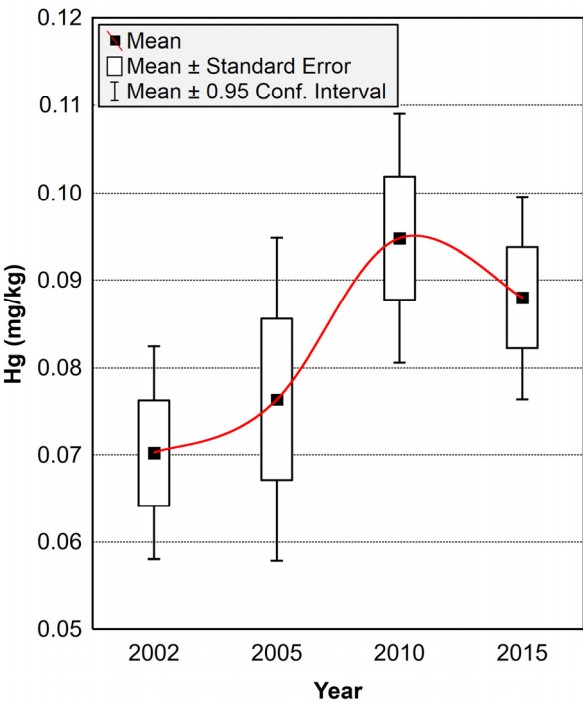

**Figure 3.** Box plots of Hg according to the year of sampling.

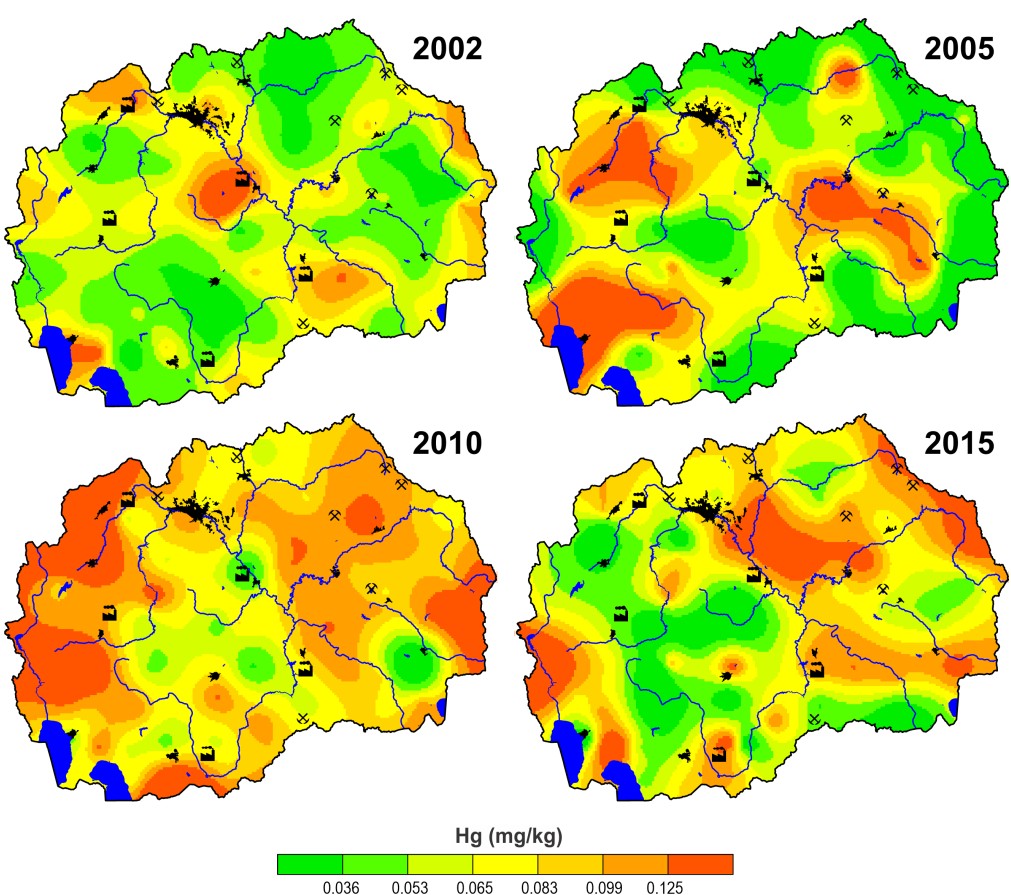

**Figure 4.** Distribution of Hg in moss samples collected in 2002, 2005, 2010 and 2015.

The mean value of mercury content in the moss samples increased in the period 2002–2010, while the value decreased for the samples collected in 2015 (Table 2, Figure 3). The coefficients of

variation (CV%) are very high, with high values of asymmetry and distribution with a large range of variation of the positively skewed concentration data indicating the influence of different natural and anthropogenic factors. In addition, an analysis of variance (ANOVA) was performed which showed significant differences between the four sampling seasons (F = 5.43/$p$ = 0.0012), which means that moss biomonitoring every 5 years is appropriate.

From the results of median values and the ranges of the content of Hg in Macedonia and those obtained in some neighbouring countries and Norway (Table 2), it can be seen that the median value obtained in 2002 (0.056 mg/kg) is slightly higher than the median value obtained from the survey in the same year in Norway [58] (which is usually considered as a pristine area [11]) (0.052 mg/kg) and 9.9 times lower than those obtained for Serbia [55]. The median value obtained for the 2005 study is comparable with the median value obtained in the similar study done in Croatia [56], lower than the median value for Slovenia [48] and 1, 5 times higher in comparison with the median value for Norway [48]. The obtained median value for 2010 is higher than the values obtained in the neighbouring countries except in the case of Albania [15], which is the case for data for 2015 as well. It should be also emphasized that the maximal value for the content of Hg in moss samples from Macedonia in 2002, 2005, and 2010 is higher than the maximal content of Hg in moss samples from Norway, while in 2015 is twice smaller.

Distribution maps based on the content of Hg in moss samples collected in 2002, 2005, 2010 and 2015 (Figure 4) show some differences and a discrepancy, i.e., variation with time as well as geographically, but it can be noticed that an increased Hg content is visible in the same regions for the whole period of the research. In 2002, the higher Hg content in the moss samples can be seen along the valley of the river Vardar, which extends from the cities of Skopje to Veles and continues southeast toward the border with Greece. It should be also mentioned that in 2002 pollution with Hg is connected with the activities of the lead and zinc smelter plant in the town of Veles [25] which had operated until 2003, as well with the work of the steelwork in Skopje [17,45]. Higher content of Hg in the mosses collected in 2002 sampling campaign was observed on the north–eastern part of the country (in the vicinity of the town of Kočani) where three Pb–Zn mines and flotation plants (Sasa, Toranica, Zletovo) are located [39].

In 2005 the enrichment of Hg was connected with the operation of the steelwork in Skopje [18] as well as the pollution by the lead and zinc smelter plant in Veles despite its closure in 2003. In contrast with the moss survey in 2002, enrichment with Hg in 2005 is also observed in the areas of Bitola in the south–western part of the country and in the west–central part of the country near the town of Kičevo, where two thermoelectric power plants using not pre–treated i.e. pre–washed prior combustion coal–lignite for energy production are located. Higher content of Hg in the mosses is also noticed in the eastern part of the country, near the town of Radoviš where copper mine and flotation were reactivated in this period.

With the reactivation of Pb–Zn mines (Toranica, Zletovo, and Sasa), in 2010 the enrichment of the content of Hg was observed in the eastern part of Macedonia. In the period of 2001/02 until 2006/07 these mines were not active, but in 2006 (Sasa) and 2007 (Toranica, Zletovo) were reactivated. In the mining area, millions of tons of flotation tailings are deposited on the ground, and can easily be dispersed into the biosphere by wind [39–41]. Furthermore, the enrichment in 2010 is also observed in the area of the steelwork in Skopje, near Bitola, Kičevo, but also near Kavadarci which is related to reactivation and the increasing of the production capacity of the ferro–nickel smelter in 2004 [19]. Higher content of Hg in the mosses collected near the western border of the country can be explained by the transboundary transport from Albania where the content of Hg in the mosses in this period was substantially higher than the samples collected in Macedonia [57].

Although the median value of Hg content in the mosses in Macedonia in 2015 is lower than the one in 2010, the high values are observed in the same areas as in the previous years.

High enrichment of the moss samples collected near the capital of Skopje, is partially connected with the presence of mercury–contaminated sites at the former "Organic Chemical Industry in Skopje

–OHIS" chlor–alkali plant situated in the town (operational from 1964 until 1995). The plant was based on mercury cell electrolysis and during the whole operational period, sodium hydroxide and hydrochloric acid were produced. It is assumed that part of the waste arising from the chlorine production containing mercury was dumped onto the mixed waste dump within the plant surrounding and that 400 tons of mercury were lost in the environment [59].

The results from this study and from previous work [24–27], in which the content of Hg in various environmental samples (soil, water, sediments) were determined, has been used to prepare the Minamata Initial Assessment in the Republic of Macedonia and the National inventory on mercury releases developed in 2017 [59]. The reference base year is 2013 and data for this year have been used in the inventory when available. The National mercury inventory identified most of the sources of mercury releases in the country and estimates or quantifies the releases which are consistent with the results obtained from these studies using moss biomonitoring.

## 4. Conclusions

The content of mercury was determined in moss samples collected in 2002, 2005, 2010, and 2015 all over the territory of Macedonia by using CV–AAS. ANOVA analysis of the results showed significant difference between the results obtained in the four sampling seasons. Although distribution maps show some differences and a discrepancy, the largest anthropogenic impact of air pollution with mercury was found near the abandoned lead–zinc smelter in the town of Veles, lead and zinc mines Sasa, Zletovo and Toranica – in the north–eastern part of the country, ferronickel smelter at the vicinity of the town of Kavadarci, and the copper mine and flotation near the town of Radoviš. The high content of mercury in the moss samples was also observed near the two thermoelectric power plants located in the vicinity of the towns of Bitola and Kičevo. Evidence of transboundary transport from Albania was observed in the western part of the country in samples collected in 2010 and 2015. Elevated values of Hg content in the moss samples collected near Skopje can be explained with the work of the former chlor–alkali plant "OHIS" and the inappropriate deposition of mercury–contaminated waste in the vicinity of this plant. This work is essential for modeling the mercury pollution in Macedonia, as well as monitoring the future trends aiming to preserve the quality of the ecosystems from deteriorating.

**Author Contributions:** Conceptualization, L.B. and T.S.; methodology, T.S.; software, R.Š.; investigation, T.S.; L.B. and K.B.A.; data curation, R.Š.; writing—original draft preparation, L.B.; writing—review and editing, L.B. and T.S.; visualization, R.Š. All authors have read and agreed to the published version of the manuscript.

**Funding:** This research received no external funding.

**Conflicts of Interest:** The authors declare no conflict of interest.

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
