# Peer review of "Atmospheric Mercury Deposition in Macedonia from 2002 to 2015 Determined Using the Moss Biomonitoring Technique"

_atmosphere, doi:10.3390/atmos11121379_

Round 1
Reviewer 1 Report
Dear authors
Please take into consideration the comments and corrections in the attached file.
Kind regards

Author Response
"Please see the attachment."
We thank the reviewer for his comments on improving the manuscript.

Reviewer 2 Report
This is a review report for the manuscript entitled
Atmospheric Mercury Deposition in Macedonia from 2002 to 2015 Studied by Moss Biomonitoring Technique by Stafilov et al.
The manuscript reports measurement results of mercury concentration in moss found over the field of Macedonia in order to find temporal deposition trend, distinguish natural mercury sources from anthropogenic sources. Moss sampling was conducted at 72 locations in the summer of 2002, 2005, 2010, and 2015. The samples were digested using microwave digestion, and then mercury content was analyzed by CV-AAS.
Overall, the manuscript was fairly well organized, but I did not find much of originality in this manuscript, unfortunately. The authors have published similar papers for other metal elements (i.e., Barandovski et al., Atmos., 2020; Stafilov et al., Air Qual. Atmos. Health, 2018), and the data were analyzed in depth in those publications. Compared to those, I was impressed that the analysis was not done in depth. To be specific, “Results and discussion”, the heart of research paper, has 2 pages, one page of which (page 5) was for the report of the observed facts, which seem usual values, and another was statements of published results mostly. Furthermore, I understood that the mercury distribution plots likely indicate that moss sampling every 5 years may not appropriate for better understanding atmospheric deposit of mercury (see my point on Figure 4 below in detail). Nevertheless, the jobs done would be a lot, and the results should be published someway, possibly combining with different type of data or analysis. To fit the aim of this journal the authors should explain what was done for Figure 4, which is the main point of this manuscript.
General comment
Many discussions on the emission sources were for specific locations, such as cities of Bitola, Kicevo, Skopje, and Veles, the valley of river Vardar, and etc. Unfortunately, I do not know where those locations are, therefore, could not follow the discussion, particularly page 6. Readers would appreciate if those locations are clearly indicated in Figure 1.
There is a nice mercury distribution map (Figure 4), but I wonder how the authors could create such high resolution distribution plots based on only 72 sampling points. Even Harmens et al. (2015), who conducted moss biomonitoring at more than 4000 locations in Europe, do not show such a high resolution distribution map. Is this really based on observations (stated in the first sentence in the second paragraph in page 5)? What is the grid size? If the plot were results of modeling, the authors then should discussed discrepancies between the modeled and observed values. The method used for making this plot should be clearly stated. Another important point in this figure is that the variation of mercury deposit is large over time and some locations show that the mercury deposits substantially decreased over time, implying that deposit mercury ran off or evaded. That is, as natural passive sampling moss biomonitoring every 2 or 5 years may not be appropriate for monitoring the deposit of atmospheric mercury.
Specific comments
The second line of the second paragraph in page 5: “Figure 3” seems a mistake, and it is supposed to mean “Figure 4” actually. Please check.
The first line (top sentence) of page 6: Please rephrase “diversification of….”. I do not understand the point.
Figure 2: I understood that Macedonia can be divided into 6 regions geologically, but I do not understand how such information was used in the interpretation. What is the relation to the mercury observations?
Figure 1 and Figure 3: The authors should combine this figure and Figure 1. Also, the inset of Figure 1, a map of Europe, is too small. Readers will appreciate to make this map bigger.
Some references (24 (not 247!) and 26) were not written in other language. Please indicate it.
Author Response

(The authors gave the same response as above.)

Reviewer 3 Report
My comments are indicated in the annotated MS. The MM needs major revision and extensive editing of English

Author Response

(The authors gave the same response as above.)

Round 2
Reviewer 3 Report
All the comments are present in the annotated text. Some of the changes requested in the first version were done by AA, but the majority were not accomplished. Moreover the MS needs extensive English revision

Author Response
To
Ms. Marijana Mandić, M.Sc.
Assistant Editor, MDPI AG
MDPI Open Access Publishing Romania SRL
Atmosphere Editorial Office
Str Avram Iancu 454, 407280, Floresti, Cluj, Romania
Dear Mr. Andrei-Cosmin Diaconu,
December 16, 2020
Thank you very much for your e-mail of December 14, 2020, with Reviewer’s comments for our manuscript " Atmospheric mercury deposition in Macedonia from 2002 to 2015 studied
by moss biomonitoring technique" (Manuscript ID atmosphere-1005825). We are sending the reviewed version accordingly to the reviewer’s comments and suggestions.
GENERAL REMARKS
NOTE: Please correct the references' numerical order in the main text:
Wrong reference number: [16], prev number is [12]
Wrong reference number: [13], prev number is [16]
Wrong reference number: [17], prev number is [15]
Wrong reference number: [48], prev number is [46]
Wrong reference number: [58], prev number is [56]
Wrong reference number: [57], prev number is [60]
Remark accepted and necessary changes are done
Reference 47 has no citation.
Remark accepted and necessary changes are done
Also, there are more than 20 refs with just one author name. Please check if
the author name part is complete.
Remark accepted and necessary changes are done
Reviewer #3
(Please take into consideration the comments and corrections in the attached file):
Remarks:
Page 5,
Fig 3. Box plots of Hg according to seasons of sampling
The remark was accepted and changed to
Fig 3. Box plots of Hg according to year of sampling
Page 5 Paragraph 3 after figure 3
Significantly or not?? use an appropriate test
The remark was accepted and additional explanation in the text was added including data obtained from the ANOVA test:
"In addition, an analysis of variance (ANOVA) was performed which showed significant differences between the four sampling seasons (F = 5.43 / p = 0.0012), which means that moss biomonitoring every 5 years is appropriate"
Page 6 First paragraph
Results are reported without a statistical support. The observed differences are based on personal interpretation. It is advisable to enrich and support the observations through the use of statistical tests. For example (but this should be considered for each comparison), in relation to the box-plots with the Hg values belonging from different “seasons” or better years, it could be appropriate to use an ANOVA test (parametric or not depending on the distribution of the data)
As it seems now clearly visible from box plots, all the values recorded are not significantly different among the various surveys, given the high variance. But it is also possible that a different arrangement of data (not all together) could put in evidence differences among specific surveys or areas of sampling.
The use of words as "slightly higher" or "higher" has no sense, the AA should put in evidence if the data are or not significantly different in the different years or in different areas of Macedonia, otherwise the survey is meaningless.
All the results and discussion need to be re-written following this concept
Corrected. Explanation in section 2.4. Statistical Methods
Page 6 Paragraph 2 Line 2 (Fig 4)
In the previous revision I asked to clarify on which data the map was built, but no additional information is given in the present version
I also asked to add on the maps the main sources hypothesized by the AA (industries, smelters, mines) to improve the interpretation of the figure
Figure 4 has been corrected. Explanation also in section 2.4. Statistical Methods
Page 6 Paragraph 3 Line 1:
All the comparison of the following 4 paragraphs are not connected to a clear data base; in other words,1) how did the data were aggregated or treated to put in evidence territorial differences, if any; 2) were these differences significant? based on which test?
Statistically, we cannot determine the difference between the distribution in Macedonia and other countries because we do not have their analyzes with we could perform the ANOVA test.
Page 6 Paragraph 3 last senetence:
Add a statistical significance; based on which data?
The ANOVA test is unreliable for groups of less than 50 samples, so we cannot statistically prove the difference between regions. Our claims are experiential and empirical.
Page 7 FIG 4
See the comments in the text to improve the Fig 4
Figure 4 has been corrected.
Page 7, Conclusion
The Conclusions should be changed in light of the above comments
Accepted and revised
Finally, the AA did not provide the extensive revision of English I recommended
The remark was accepted and the extensive revision of English.
Academic Editor Notes
Please make sure that you have addressed all the reviewers concerns and edit the paper to improve English (sentence structure) where possible.
For example, consider the following changes:
“Atmospheric mercury deposition in Macedonia from 2002 to 2015 studied by moss biomonitoring technique” to “Atmospheric mercury deposition in Macedonia from 2002 to 2015 determined using moss biomonitoring”
The remark was accepted.
“Mercury distribution maps provide the sites with increased content of mercury in moss samples mostly as a result of anthropogenic pollution” to “Mercury distribution maps show that sites with increased concentrations of mercury in moss are likely impacted by anthropogenic pollution”
The remark was accepted.
“The results were compared with the results obtained from similar studies done in the mentioned years in the neighboring counties and in Norway—which is a pristine area and serves as a reference, and it was concluded that mercury air pollution in Macedonia is significant only in industrialized regions” to
“The results were compared similar studies done during the same years in neighboring countries and in Norway—which is a pristine area and serves as a reference, and it was concluded that mercury air pollution in Macedonia is significant primarily in industrialized regions”
The remark was accepted.
“known anthropogenic activities in this regions,” to “known anthropogenic activities in the region”
The remark was accepted.
“Detailed description of the country (location, climate and demographics) can be found in [29,30] as well as the previously works [17-23].” to “A detailed description of the country (location, climate and demographics) can be found elsewhere [17-23, 29,30].
The remark was accepted.
Also consider removing: “Obtained results were statistically analyzed using descriptive statistics.” This goes without saying.
The remark was accepted.
We are very grateful to the reviewer for his positive opinion of the manuscript.
All changes in the manuscript are marked in red color.
We hope that this version of the manuscript will be accepted for publication in the Atmosphere.
Sincerely yours,
Trajče Stafilov and
Lambe Barandovski
